# A Comparison of Two Analytical Approaches for the Quantification of Neurofilament Light Chain, a Biomarker of Axonal Damage in Multiple Sclerosis

**DOI:** 10.3390/ijms241310787

**Published:** 2023-06-28

**Authors:** Anna Pafiti, George Krashias, John Tzartos, Socrates Tzartos, Christos Stergiou, Eftychia Gaglia, Irene Smoleski, Christina Christodoulou, Marios Pantzaris, Anastasia Lambrianides

**Affiliations:** 1Postgraduate School, The Cyprus Institute of Neurology and Genetics, Nicosia 2371, Cyprus; pafitia@cing.ac.cy (A.P.); georgek@cing.ac.cy (G.K.); cchristo@cing.ac.cy (C.C.); pantzari@cing.ac.cy (M.P.); 2Neuroimmunology Department, The Cyprus Institute of Neurology and Genetics, Nicosia 2371, Cyprus; 3Department of Molecular Virology, The Cyprus Institute of Neurology and Genetics, Nicosia 2371, Cyprus; 4B’ Neurology Department, School of Medicine, National and Kapodistrian University of Athens, “Attikon” Univeristy Hospital, 10676 Athens, Greece; jtzartos@gmail.com; 5Tzartos NeuroDiagnostics, 3, Eslin Street, 11523 Athens, Greece; stzartos@gmail.com (S.T.); cstergiou@neurodiagnostics.gr (C.S.); 6Clinical Sciences, The Cyprus Institute of Neurology and Genetics, Nicosia 2371, Cyprus; eftychia@cing.ac.cy (E.G.); irenesm@cing.ac.cy (I.S.)

**Keywords:** multiple sclerosis, neurofilament light chain, enzyme-linked immunosorbent assay, single molecular array advanced technology

## Abstract

Neurofilament light chain (NfL), is a neuron-specific cytoskeletal protein detected in extracellular fluid following axonal damage. Extensive research has focused on NfL quantification in CSF, establishing it as a prognostic biomarker of disability progression in Multiple Sclerosis (MS). Our study used a new commercially available Enzyme-Linked Immunosorbent Assay (ELISA) kit and Single Molecular Array (Simoa) advanced technology to assess serum NfL levels in MS patients and Healthy Controls (HC). Verifying the most accurate, cost-effective methodology will benefit its application in clinical settings. Blood samples were collected from 54 MS patients and 30 HC. Protocols accompanying the kits were followed. The ELISA thershold was set as 3 S.D. above the mean of the HC. For Simoa, the Z-score calculation created by Jens Kuhle’s group was applied (with permission). Samples exceeding the threshold or z-score ≥1.5 indicated subclinical disease activity. To our knowledge, this is the first study to find strong-positive correlation between ELISA and Simoa for the quantification of NfL in serum (r = 0.919). Despite the strong correlation, Simoa has better analytical sensitivity and can detect small changes in samples making it valuable in clinical settings. Further research is required to evaluate whether serum NfL quantification using ELISA could be utilized to predict disability progression.

## 1. Introduction

Multiple sclerosis (MS) is a chronic autoimmune disease of the central nervous system (CNS). Despite its unknown etiology, the main pathological hallmark of the disease is the presence of inflammatory lesions which consequently result in neuronal demyelination and variable extent of axonal damage [1]. In the CNS, oligodendrocytes produce and maintain multiple layers of myelin under normal circumstances, which act as an insulator and allow for fast nerve conduction [2]. However, in MS, the destruction of myelin causes nerve conduction to be slower and can eventually lead to the loss of nerve fibres, resulting in the breakdown of myelin, which exposes the axon [3]. The cytoskeleton of the axon is made up of three structural proteins known as neurofilament light, heavy and medium [4]. Under physiological conditions, neurofilaments are highly stable in axons, and are thought to be critical for their radial growth and stability. They enable effective and high-velocity conduction [5,6]. Specifically, neurofilament light (NfL) has been identified as a biomarker of axonal damage in various neurodegenerative diseases including MS [7]. Extensive research has been conducted on its detection and quantification, especially in cerebrospinal fluid (CSF) samples [4,8].

Currently, MS diagnosis and monitoring relies on MRI imaging, the presence of oligoclonal bands in the CSF and clinical events [9]. However, none of them have the ability to detect damage to neurons and axons, which have been strongly correlated to disease progression and long-term disability [10]. At a panel organised by the International Progressive MS Alliance, experts agreed on the importance of tissue fluid biomarkers in MS, with their focus being on neurofilament light chain (NfL) quantified from serum samples [11]. Since CSF sampling involves a more invasive procedure than obtaining a blood sample, the hope is that future biomarker studies will involve serum samples that were also been found to correlate with disease progression, with disability measures and that were able to predict response to treatment. The most widely used assay for the concentration of NfL in serum is the Quanterix platform, which uses single molecular array (Simoa) technology [11]. Commercially available Enzyme-linked immunosorbent assay (ELISA) kits offer an alternative, inexpensive approach to NfL quantification. Regardless of the technique used, samples analysed should correlate and the conclusions should be universal. An important aspect of a biomarker is that it should be able to be measured accurately and reproducibly [12].

Our current study involved a comparative study of NfL quantification in serum samples using Simoa technology (single molecule array providing increased sensitivity) and ELISA. Simoa advanced technology involves the use of magnetic beads that consist of antibodies that bind to the target of interest within the sample. Target-specific biotinylated detection antibodies and streptavidin enzyme conjugate result in an immunocomplex consisting of a bead analyte, detection antibody, and reporter enzyme. Detection of the target is indicated by a fluorescent signal from the Simoa bead technology [13]. ELISA involves the same analytes, only in this case, the capture antibody is pre-coated on a solid surface that is used to capture NfL. The detection antibody is biotin conjugated, and the addition of HRP-conjugated Streptavidin allows for quantitative determinations by enzymatic turn-over to a coloured product.

The aim of this study was to evaluate the concentration of NfL in the serum of MS patients and healthy controls (HCs) using the two analytical platforms. Identifying and validating the most accurate technique for quantifying NfL in serum will be of significant benefit in clinical settings. To our knowledge, this is the first study to compare the newly developed ELISA with the Simoa technology using serum samples from MS patients.

## 2. Results

### 2.1. Patient Demographics

The current study includes blood samples from 54 MS patients and 30 HC volunteers, randomly selected around the same time. No significant difference was found between the age/gender of patients and controls (*p* = 0.6193/0.2207, respectively). The ages ranged from 28 to 68 years old for the MS group and 31 to 68 years old for the HC group. At the time of sample collection, 52 out of 54 (96%) patients were being treated with Rituximab, while the rest (4%) were treated with Ocrelizumab. All patients had been diagnosed with Secondary Progressive Multiple Sclerosis (SPMS). Patient demographics are reported in Table 1.

### 2.2. Quantification of NfL Concentration Using ELISA

NfL concentration was determined using the calibrator curve and by correlating the absorbance to the known concentration. The threshold was determined as 3 S.D. above the mean of the HCs; concentrations above the threshold can be considered abnormal and may potentially have ongoing disease activity. MS patients had a significant higher NfL concentration than HC (*p* = 0.0006) (Figure 1). None of the HC participants had a concentration above this threshold. A total of 11 MS patients were found to have abnormal levels of NfL. MS patients had a higher mean concentration of 11.61 pg/mL [Interquartile range from 5.286 to 15.78] in comparison to HCs with a mean of 6.640 pg/mL [3.917 to 8.805].

### 2.3. Quantification of NfL Concentration Using Simoa Technology

Identical serum samples from MS patients and HCs were analysed using Simoa technology. Likewise, a significant difference was seen between MS patients and HC groups; however, on this occasion, the significance was *p* = 0.0014 (Figure 2). Higher mean concentrations, 14.38 pg/mL [8.908 to 18.63] were observed in the MS group than in the HC group, 9.859 pg/mL [7.140 to 11.73]. A calculation developed by the Jens Kuhle group was used to calculate the z-scores and percentiles for each sample, where a z-score above or equal to 1.5 can be associated with an increased risk of future clinical activity [14]. Figure 3 shows the z-scores in comparison to the NfL concentration. Sixteen patients with MS had z-scores of above 1.5, whereas three HCs had z-scores above the threshold. The calculation takes into consideration the subject’s BMI and age. NfL concentration is expected to increase with age and decrease with BMI [14]. Figure 4 and Figure 5 illustrate that our participants followed the expected pattern. A threshold of 3 S.D. above the mean of the HCs was also employed here. 

### 2.4. Comparison of the Two Analytical Methodologies for the Detection/Quantification of NfL in Serum

Both methodologies yielded similar results in terms of where each sample lies in comparison with the rest of the samples analysed. Following Spearman’s correlation, whereby the ranking of values was taken into consideration, the r-value was 0.8691. However, the concentration values were found to be significantly different for both groups (MS, *p* < 0.0001; HC, *p* < 0.0001) (Figure 6). A line links the same serum sample to show a clearer comparison of the concentration obtained following the two different methodologies using the same sample. For example, the serum sample ‘MS30’ had the highest concentration according to both the ELISA and Simoa methods. The three samples with the highest concentrations were the same samples following both methodologies. An additional correlation test, Pearson’s correlation, was performed to compare the two methodologies (Figure 7), where an R^2^ value of 0.8445 was obtained, indicating a strong positive correlation. Finally, in Figure 8, the samples that were found to have a concentration considered abnormal in both methodologies are shown. All but five of those samples were found to have abnormal concentrations using both methodologies, since they were either above the threshold that we set or had sufficiently high z-scores.

## 3. Discussion

To our knowledge, two other studies have compared methodologies for the quantification of NfL in serum [15,16]. Specifically, ELISA and Simoa were compared in a study by Jens Kuhle et al.; however, at the time, the Simoa method was still under development and the ELISA used had an analytical sensitivity of 78.0 pg/mL (as well as being advised to be used with CSF samples, not serum) [15]. Revendova et al. also used an ELISA kit by Uman Diagnostics, with analytical sensitivity of 4 pg/mL [16]. In our study, the most updated version of the Simoa method and NfL advantage kit were used, with analytical sensitivity of 0.174 pg/mL and dynamic range of 1800 pg/mL in serum. While the ELISA had an analytical sensitivity of 0.8 pg/mL and dynamic range of 4–100 pg/mL for the use of serum samples. In view of these advances, it is certain that an updated evaluation of the two methodologies is required.

Importantly, our study has demonstrated that there is a strong correlation between the concentrations of NfL in serum obtained after comparing the two methodologies, with an R-value of 0.9190, and *p* < 0.0001, as shown in Figure 7. In contrast, Jens Kuhle et al. showed a much weaker correlation (r = 0.43 and a *p*-value of 0.013) [15]. Likewise, Revendova et al. presented significant differences between the two methods, concluding to an r-value of 0.543 and a *p*-value of 0.001 [16]. Nevertheless, further studies are required to determine the correlation between serum levels obtained using the newly available ELISA kit and CSF levels. Based on our results, given that NfL concentrations in serum using ELISA correlated to those obtained using the Simoa method, we can only assume that correlation will be strongly positive, since other studies have shown that Simoa serum levels are highly correlated to CSF [13]. Furthermore, Figure 6 shows the comparison of the samples analysed with the two methodologies ranking in a similar order, where a strong positive correlation was found (r = 0.8691).

The establishment of an age- and BMI-adjusted model by the Kuhle group added a huge benefit to the results obtained using the Simoa method [14]. Serum NfL levels can be a fairly unstable measure, as they increase with age and decrease with BMI. Consequently, in a group-level comparison, this limits the ability of serum NfL to act as a reliable biomarker [14]. To demonstrate this using our data, Pearson correlation analysis was applied (Figure 4 and Figure 5). Positive correlation between age and NfL concentration, using the Simoa method, was established in both study groups, MS and HC (r = 0.447/0.607 respectively). In the case of BMI, due to the small sample size, the expected negative correlation was only seen in the MS patients (r = −0.125). A stronger correlation would be expected with a larger sample size as well as a better distributed BMI among the participants.

With permission from the authors of the age- and BMI-adjusted model, we applied the internet-based app and calculated the z-scores for our participants. According to Pascal Benkert et al. from the analysis performed on a large cohort of HCs, a z-score above 1.5 is considered to have an increased risk in future clinical activity [14]. This refers to relapsing, Expanded Disability Status Scale (EDSS) worsening or EDA-3 [14]. Future research should include longitudinal samples and monitoring of patients with a z-score above or equal to 1.5 and analyze whether they had shown any signs of disability worsening. Figure 3 shows the z-scores compared to the concentrations of NfL using the Simoa method. The dashed line indicates a threshold of 1.5, where any samples above the threshold were considered to indicate ongoing disease activity. However, such a calculation is not available for NfL concentrations obtained with ELISA. For this reason, we introduced a threshold for ELISA that we and others have applied previously during a patient-control cohort analysis of other serum/CSF markers [17,18,19]. The threshold was calculated as 3 S.D. above the mean of the HC. Of note, as shown in Figure 8, 11/16 participants who had a z-score above or equal to 1.5 also had an NfL concentration above the threshold according to the ELISA methodology. The likely explanation for the five MS patients who had a z-score above 1.5 but below the threshold according to the ELISA test, could be that they either have a high BMI or are younger than what is to be expected based on their NfL levels (presented in detail in Table 2). It is important to note that concentrations followed the same pattern. For instance, MS10 had an NfL concentration below the mean following both methodologies. Although this particular patient had a low NfL concentration, the young age together with the high BMI, correlated with ongoing disease activity. Since the ELISA threshold is solely based on concentration values, it does not account for such variables that affect NfL concentrations. For comparison purposes, the same threshold, 3 S.D. above the mean of the HCs, was also applied using the Simoa technology data. As can be seen in Figure 2, no HCs had a concentration above the threshold. The same applies to the ELISA data (Figure 1). In regards to the MS patients, 8 out of the 11 patients with an NfL concentration above the ELISA threshold had a concentration higher than the Simoa threshold. None of the five patients presented in Table 2 had a Simoa concentration above the threshold. Despite an obvious overlapping in the trends seen, we suggest the use of the z-scores for Simoa instead of the threshold. The fact that they take into account the variables that affect NfL concentration will result in more reliable conclusions regarding whether the NfL levels are due to axonal damage from ongoing disease activity [14]. 

One limitation of our study is that the z-score calculation is based on the concentrations of Swiss national HCs [14]. There are no data available on the healthy control concentrations of the Greek-Cypriot population, where our study participants were from. To our knowledge, this is the first study evaluating NfL levels in Greek-Cypriots. Further studies are required to establish the normal NfL ranges in different populations. Once these data are available, we can re-evaluate our results to determine whether there is any significant difference in the conclusions drawn. Additionally, we have no data on whether our patients at the time of sample collection had ongoing disease activity, which could alter the level of NfL. However, MS patients treated with B-cell depleted agents such as Rituximab show stable NfL levels, absence of clinical relapses and no detectable MRI activity [20]. All of the patients in this study were treated with B-cell-depleting medication. 

The NfL concentrations for both MS patients and HCs using the two methodologies were found to be significantly different. This could be due to differences in the analytical sensitivity of the methods applied [15]. The Simoa method has an analytical sensitivity of 0.174 pg/mL, while the ELISA method has an analytical sensitivity of 0.8 pg/mL; therefore, the ELISA method can detect more neurofilaments. Hence, the higher mean observed. This is an important point to consider when comparing longitudinal samples from the same patient. It can be assumed that minor changes may be more easily detected using the Simoa method, than the ELISA method. Further analysis with longitudinal samples is required to support the hypothesis. Nevertheless, ELISA is more likely to provide accurate results in patients with high concentrations of NfL. Furthermore, ELISA has inter- and intra-plate variability. In our study, we allowed for <10% interplate variability in the standards before assessing the concentration of the samples. Additionally, all samples were assayed in duplicate. 

In conclusion, this is the first study to show a strong positive correlation between Simoa Advanced technology and ELISA in the quantification of serum NfL as a measure of axonal injury. The Simoa method has been granted Breakthrough Device designation by the U.S. Food and Drug Administration (FDA) as a prognostic aid in assessing the risk of disease activity in MS patients [21]. Additionaly, Simoa technology has more advantages in evaluating NfL in patients within clinical settings. Because of the increased analytical sensitivity, even minor changes in longitudinal samples will be detected. Studies have shown that CSF and serum NfL concentration in MS patients can predict the course of progression of the disease and forecast upcoming MRI scans [11]. This will allow for faster pharmaceutical intervention before a relapse occurs. NfL concentration can also assess the efficacy of medication. Nig Liu et al. showed a significant decrease in NfL concentration after therapy with natalizumab. Thus, NfL measurements proved the success of natalizumab in decreasing neurodegeneration [22].

The strong correlation that we have shown in the two analytical approaches clearly point to the reliability of using ELISA for research purposes. Laboratories that do not have access to the Simoa method can still conduct their research to produce reliable and accurate results for the quantification of NfL in serum. We must take into consideration the price difference betwenn the two methods. In our case, it cost us 10€ for ELISA and 100€ for Simoa per sample. Additionally, the Simoa method requires the purchase of an HD-X SIMOA bead-based immunoassay analyzer. 

Our results support the use of serum NfL measurements and demonstrate that serum NfL can be consistently detected on various suitably sensitive assay platforms. To further support the use of NfL as a biomarker of axonal injury, longitudinal analysis needs to be performed. NfL levels will need to be correlated to EDSS scores and new clinical episodes, as well as MRI scans that show the progression of the disease. 

## 4. Materials and Methods

### 4.1. Study Design and Participants

The study included blood samples from 54 participants diagnosed with MS, according to the McDonald criteria, at the Cyprus Institute of Neurology and Genetics. Inclusion criteria were as follows: above the age of 18; demographic and clinical data easily available; clearly defined clinical course; ability to provide informed consent along with their Body Mass Index (BMI). Exclusion criteria were as follows: inability to provide consent; diagnosis with any other neurodegenerative disorders. Additionally, blood samples were collected from 30 control participants with no history of any neurodegenerative disorders, again providing informed consent. Patient samples and controls were age- and sex-matched.

### 4.2. Blood Processing

Blood samples were collected in BD Vacutainer red-top plain tubes containing clotting activators at the Cyprus Institute of Neurology and Genetics. The samples were allowed to clot at room temperature and centrifuged at 1500× *g* for 10 min. Serum was extracted and stored in aliquots of up to 1 mL at −20 °C.

### 4.3. Quantification of NfL Concentration

#### 4.3.1. NfL Antibody Detection Using Sandwich ELISA

A commercially available ELISA kit (Ref. Number; 20-8002 ROU, Uman Diagnostic, Quanterix, Umea, Sweden) was used for the quantification of NfL in serum. The protocol accompanying the kit was followed. Briefly, diluted serum from MS patients, healthy controls and calibrators were added to a 96-well plate and coated with a non-competitive, monoclonal antibody that allowed binding of NfL. Following a two-hour incubation, the biotin conjugated detection antibody was added and the plate was kept at room temperature for 90 min to allow for binding to take place. Streptavidin horseradish peroxidase-conjugate followed after the incubation. The plate was washed after each incubation using the wash buffer provided. The conjugate was bound to biotin and the enzymatic reaction was then detected through the addition of TMB substrate. The reaction was stopped using diluted H_2_SO_4_, and absorbance was immediately measured using an absorbance reader. The calibrator curve was then used to correlate the absorbance to the concentration of NfL. 

#### 4.3.2. NfL Antibody Detection Using Simoa

The NF-light Advantage Assay Kit (Quanterix, Billerica, MA, USA) was used for quantitative determination of NfL in serum with an HD-X SIMOA bead-based immunoassay analyzer. Samples were tested in duplicate and the NfL percentiles and z-scores were calculated using the internet-based application available at https://shiny.dkfbasel.ch/baselnflreference (accessed on 30 June 2022) of Jens Kuhle group, with kind permission [14], where each value was compared with healthy controls of similar age and BMI.

### 4.4. Statistical Analysis

Statistical analysis was carried out using GraphPad Prism V.8.0.1 for Windows (La Jolla, CA, USA). An unpaired *t*-test was done to assess any significant differences between the concentration of NfL in MS patients and healthy controls, age and sex. Paired *t*-test was done for the comparison of NfL concentration between the two methodologies. Significance was set at *p* < 0.05. Additionally, correlation of the concentration of each sample and their ranking following the two analytical procedures was assessed.

## Figures and Tables

**Figure 1 ijms-24-10787-f001:**
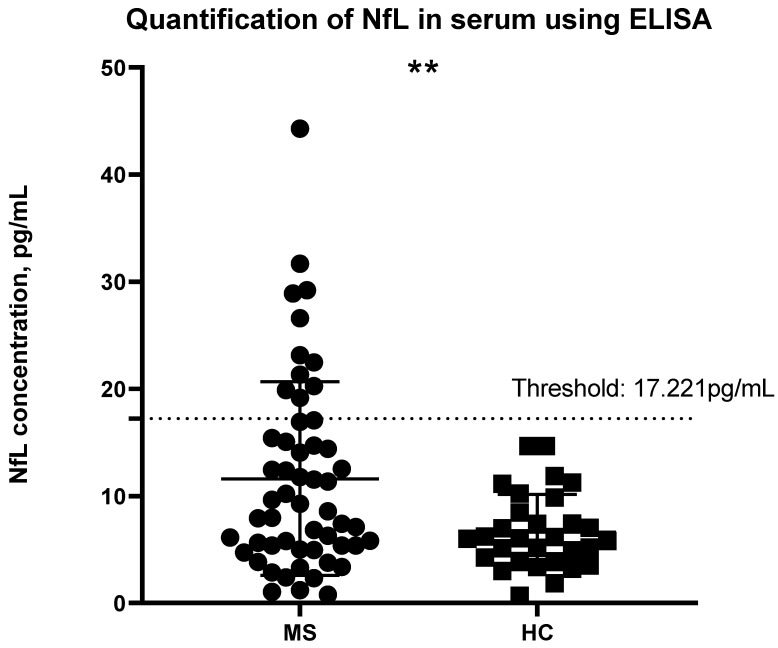
Quantification of NfL concentration in the serum of MS patients and HCs using ELISA. The threshold has been set as 3 S.D. above the mean of the HCs, indicated by the dashed line. NfL concentration above the threshold is associated with an increased risk in clinical activity. Welch’s *t*-test was applied for the statistical analysis, significant difference was found between the two groups (** *p* < 0.008). MS: Multiple Sclerosis; HC: Healthy Controls.

**Figure 2 ijms-24-10787-f002:**
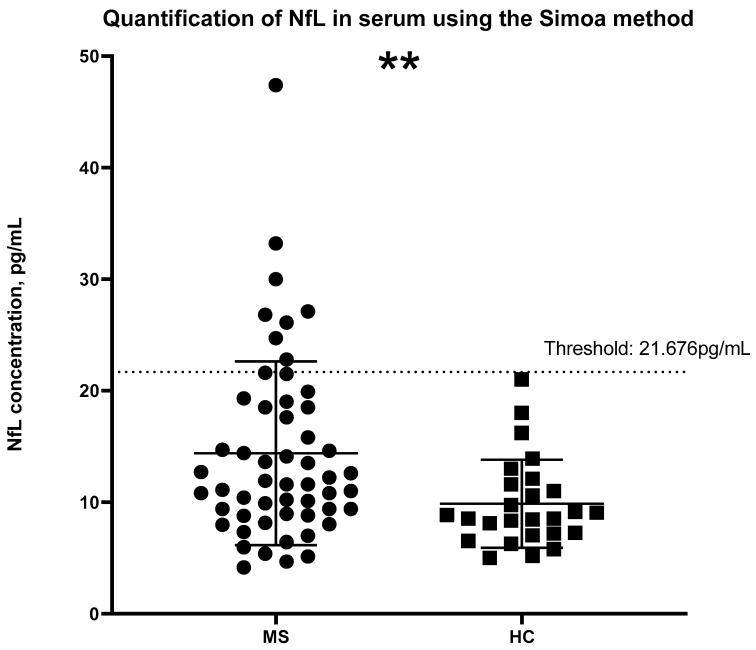
The Simoa method was applied to analyse NfL concentration in serum samples of MS patients and HCs. Welch’s *t*-test was applied for the statistical analysis, significant difference was found between the groups with a (** *p* < 0.001). For comparison purposes, the threshold has been set as 3 S.D. above the mean of the HCs and is indicated by the dashed line. MS: Multiple Sclerosis; HC: Healthy Controls.

**Figure 3 ijms-24-10787-f003:**
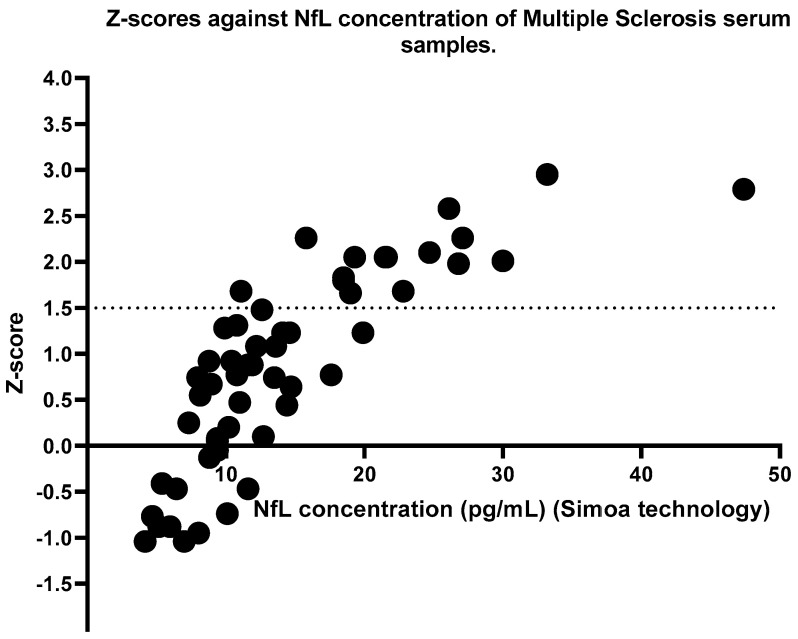
Z-scores were calculated using the internet-based application created by the Jens Kuhle group (https://shiny.dkfbasel.ch/baselnflreference (accessed on 30 June 2022)), where the NfL concentration from the Simoa method together with the participant’s BMI and age are taken into consideration. A z-score more or equal to 1.5 is considered abnormal and is indicated by the dashed line. Sixteen MS patients had a z-score above 1.5. MS: Multiple Sclerosis; BMI = Body Mass Index.

**Figure 4 ijms-24-10787-f004:**
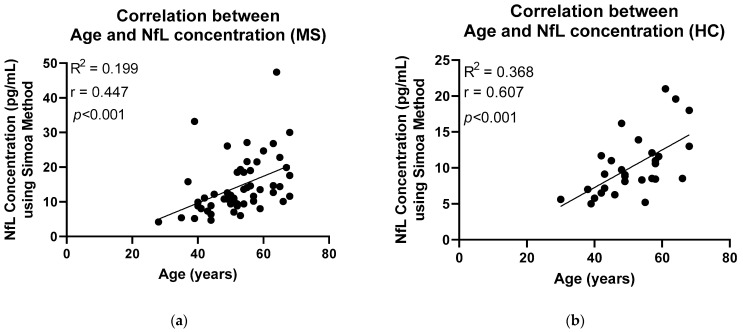
Pearson correlation has been applied to illustrate the relationship between age and NfL concentration using the Simoa method. (**a**) A moderate positive correlation (r = 0.447) has been found between age and NfL concentration of the MS patients. (**b**) A moderate positive correlation (r = 0.607) has been found between age and NfL concentration of the HC participants. MS: Multiple Sclerosis; HC: Healthy Controls.

**Figure 5 ijms-24-10787-f005:**
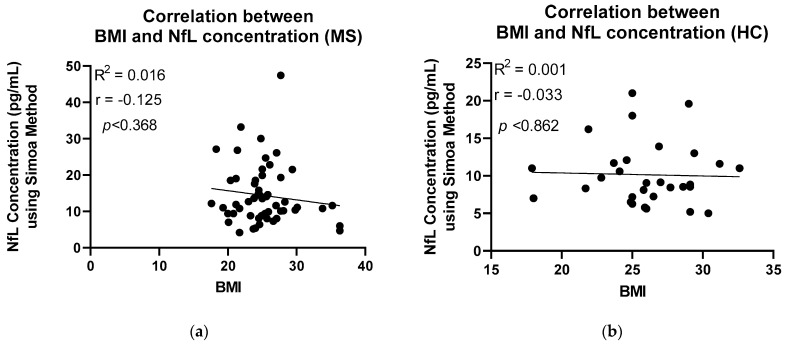
The relationship between BMI and NfL concentration using the Simoa method was analysed using Pearson correlation. (**a**) A very weak negative correlation (r = −0.125) has been found between BMI and NfL concentration of the MS patients. (**b**) No correlation (r = −0.033) has been found between BMI and NfL concentration of the HC participants. MS: Multiple Sclerosis; HC: Healthy Controls; BMI: Body Mass Index.

**Figure 6 ijms-24-10787-f006:**
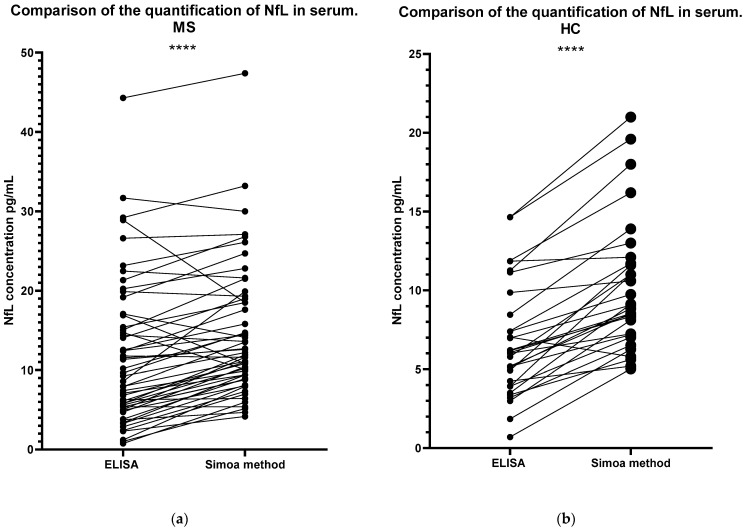
Comparison of the Simoa method and ELISA for the quantification of NfL concentration in the serum of MS patients (**a**) and HCs (**b**). For a clear comparison, a line has linked the same serum samples. A strong positive correlation has been found in the ranking of the samples in the two techniques (r = 0.8691), utilizing Spearman’s rank correlation. Paired *t*-test was used to analyse the significant difference in the NfL concentrations quantified using the two methodologies. Significant difference was found for both the MS group and for the HC group (**** *p* < 0.0001). MS: Multiple Sclerosis; HC: Healthy Controls.

**Figure 7 ijms-24-10787-f007:**
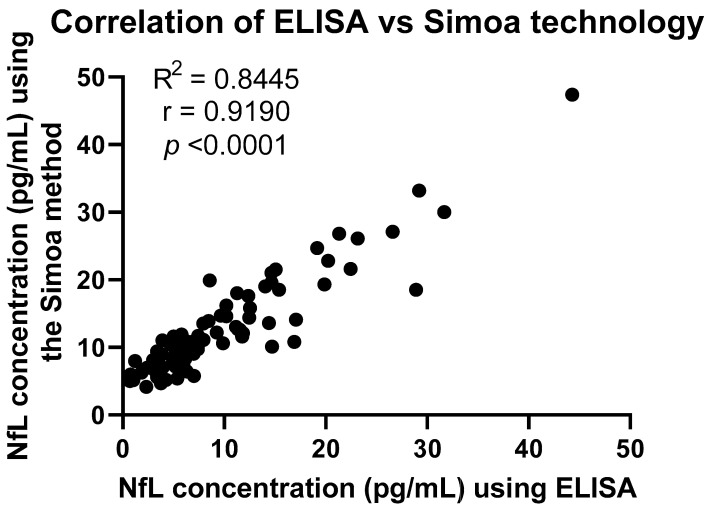
A positive correlation was found between the Simoa method and ELISA with an R^2^ value of 0.8445 applying Pearson’s r.

**Figure 8 ijms-24-10787-f008:**
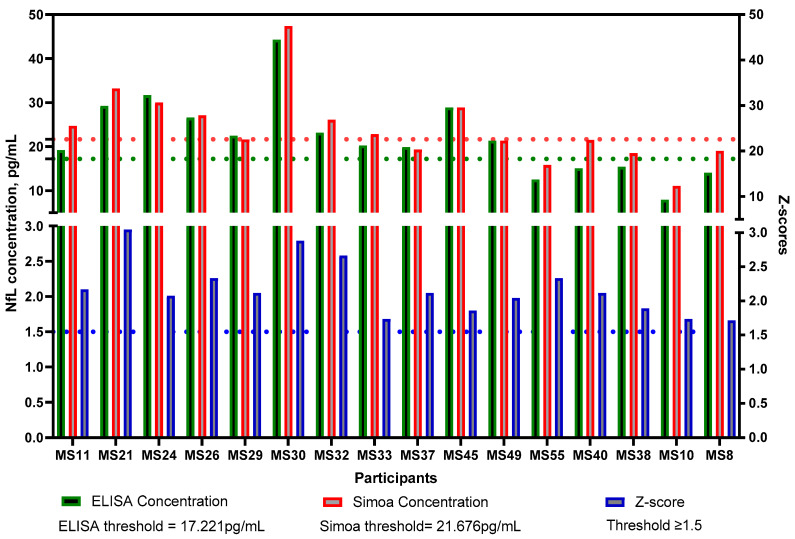
All 16 MS patients with a Z-score above the threshold of 1.5 (indicated by the blue dashed line). In comparison, all but 5/16 for ELISA and 8/16 for Simoa, had a concentration above the threshold that has been set as 3 S.D. above the mean of the HC (indicated by the green and red dashed line). MS: Multiple Sclerosis; HC: Healthy Controls.

**Table 1 ijms-24-10787-t001:** Patient Demographics.

	MS ^1^	HC ^2^
No. of Subjects	54	30
Male/Female	23/31	13/17
Mean age (±S.D. ^3^)	52.25 (±9.695)	51.17 (±9.037)
Disease Status	SPMS ^4^ (100%)	N/A ^5^
Medication	Rituximab (96%) Ocrelizumab (4%)	N/A

^1^ MS, Multiple Sclerosis; ^2^ HC, Healthy Controls; ^3^ S.D., Standard Deviation; ^4^ SPMS, Secondary Progressive Multiple Sclerosis; ^5^ N/A, Not applicable.

**Table 2 ijms-24-10787-t002:** Details of the five MS patients with a z-score above the threshold but not above the ELISA or Simoa technology thresholds.

Lab Number	BMI ^1^	Age	Simoa Concentration (pg/mL)	ELISA Concentration (pg/mL)
MS8	21.23	56	19 (↑ ^2^)	14.05 (↑)
MS10	30.10	43	11.1 (↓ ^3^)	7.96 (↓)
MS38	24.02	54	18.5 (↑)	15.40 (↑)
MS40	29.41	59	21.5 (↑)	15.06 (↑)
MS55	24.48	37	15.8 (↑)	12.53 (↑)

^1^ BMI, Body Mass Index; ^2^ ↑ above the mean; ^3^ ↓ below the mean.

## Data Availability

The raw data of the study are available upon reasonable request to the corresponding author.

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
