# Peer review of "A Comparison of Two Analytical Approaches for the Quantification of Neurofilament Light Chain, a Biomarker of Axonal Damage in Multiple Sclerosis"

_ijms, 2023, doi:10.3390/ijms241310787_

Round 1

Reviewer 1 Report

Major:

The authors have compared two analytical methodologies ELISA and Simoa for the quantification of NfL in serum of MS-patients and healthy control. For the NfL concentration obtained by two methods in the same patients, in one case, the authors use a threshold in units of concentration according to the NfL level in healthy people (average from healthy + 3SD), in the other case, a dimensionless Z-scores value is used to set the threshold. At the same time, the authors say that “such a calculation is not available for NfL concentrations obtained with ELISA”.

Both methods correlate well with each other (data obtained by the authors, r=0.919). At the same time, the levels of NfL concentration measured by ELISA are lower compared to the Simoa method. If the same threshold is used for the Simoa method as for the ELISA (mean of healthy +3SD) would the same 11 MS patients were found to have abnormal levels of NfL as in the ELISA? And will there also be None of the HC participants who had a concentration above this threshold?

The text does not specify the dynamic range, analytical sensitivity, and reproducibility for the two methods. 

Minor:

Throughout the text:

The signatures are not clear enough in some Figure

Please check extra and missing punctuation marks (“3 S.D” - 3 S.D., “Pearson’s r. .” and others)

Page 1, line 20: “to assess NfL levels in MS patients and Healthy Controls (HC)” – In serum?

Page 2, line 66: “quantification/detection’’ – quantification as both systems are quantitative analysis

Page 2, line 81: ” a large cohort of MS patients” - 54 MS patients is not a large cohort

Figure 1.: “NfL concentration pg/ml” - NfL concentration, pg/ml

Figure 1.: “NfL concentration above the threshold is associated with an increased risk in clinical activity.” – The authors should move this statement into the text and discuss it.

Figure 1.: “11 MS 107 patients had NfL concentrations above the threshold. No HC had NfL concentrations above the 108 threshold.” - This is a repeat from the text, not needed in the figure caption.

Page 3, line 113: “Likewise, a significant difference was seen between MS patients and HC groups” I can't see a significant difference. Yes, the medians are different. But we can only speak of a trend. The values for the two groups overlap. Based on a single measurement, it is impossible to distinguish between MS and HC.

Page 3, line 116: “A calculation developed by the Jens Kuhle group was used to calculate the Z-scores“ - The authors should add reference here.

Page 4, line 118: It’s not clear what the authors mean here by “an increased risk of future clinical activity”

Page 4, line 118: “NfL concentration is expected to increase with age and decrease with BMI.” - this statement requires the addition of a corresponding reference

Figure 2: “significant difference was found between the groups with a (**p < 0.001)” – Please, check if this correct

Figure 2: Please add a threshold similarly as in Figure 1.

Figure 4. “A moderate positive correlation (r= 0.447) has been found 135 between age and NfL concentration of the MS patients “- Correlation coefficients whose magnitudes are between 0.3 and 0.5 indicate variables which have a low correlation.

Figure 5. “A weak negative correlation (r= -0.125) has been found between 140 BMI and NfL concentration of the MS patients” - Is there really any correlation?

Page 10, line 243 “This could be due to differences in the analytical sensitivity of the methods applied [15]” - Authors should add analytical sensitivity here.

Page 10, line 246: “Minor changes may be more easily detected using the Simoa method than the ELISA” - This is not shown in the study.

Page 10, line 257: “Studies have shown that NfL concentration in MS patients can predict the course of progression of the disease and forecast upcoming MRI scans [11].” – in the serum?

Page 11, line 287 “Serum was extracted and stored in aliquots of up to 1 ml at -20 ˚C” – Is there any data how freeze-thaw affects NfL concentration?

Page 11, line 291 “ELISA kit (Uman Diagnostic, Quanterix)” - catalog number, country?

Page 11, line 317 “5. Patents - delete

Author Response

Reviewer 1:

Comment 1 addresses the fact that a threshold of 3 S.D. above the mean of the HCs was used for the ELISA, while the calculation for z-scores created by Jen Kuhle’s group was used (with permission) for Simoa. Reviewer 1 raised the question ‘If the same threshold is used for the Simoa method as for the ELISA (mean of healthy +3SD) would the same 11 MS patients were found to have abnormal levels of NfL as in the ELISA? And will there also be none of the HC participants who had a concentration above this threshold?’

We have calculated the threshold for Simoa and included it in Figure 2 and 8. None of the HCs had a concentration above the threshold and 8/11 MS patients were found to have NfL levels above the threshold.

Comment 2 refers to the lack of information in the text in regards to dynamic range, analytical sensitivity and reproducibility for the two methods. The analytical sensitivity of Simoa advantage kit used is 0.174pg/ml and the dynamic range in serum is 1800pg/ml; both are now stated in page 9 line 186 and page 10 line 257. In the case of ELISA, the analytical sensitivity of the kit is 0.8pg/ml (page 9 line 188) however the manufacturer does not provide the dynamic range. Additionally, we allowed for <10% interplate variability in the standards, before assessing the concentrations and all the samples were tested in duplicate (page 10, line 264-266). 

Comments 3 to 7 mention minor changes that should be made in the text. Specifically,  ‘Please check extra and missing punctuation marks (“3 S.D” - 3 S.D., “Pearson’s r. .” and others).; Page 1, line 20: “to assess NfL levels in MS patients and Healthy Controls (HC)” – In serum.?; Page 2, line 66: “quantification/detection’’ – quantification as both systems are quantitative analysis.; Page 2, line 81: ” a large cohort of MS patients” - 54 MS patients is not a large cohort.; Figure 1.: “NfL concentration pg/ml” - NfL concentration, pg/ml’. All comments have been addressed in the text and changes are highlighted.

Comment 8 and 9, questions the significant difference between MS patients and HC groups. We agree that there is an overlap between the two groups. However, overlapping error bars does not mean that the data cannot be significantly different. No conclusions can be made by looking at Standard Deviation error bars. (https://www.graphpad.com/guides/prism/latest/statistics/stat_relationship_between_significa.htm ) The statistical analysis (p < 0.0014) is valid.

Comment 9 and 10 advises the addition of references. References have been added to both points.  Page 4, line 117 and line 121.

Comment 11 suggests that we should further clarify what we mean by “an increased risk of future clinical activity” in patients with a z-score above or equal to 1.5. We base our statements on the paper published by Jen Kuhle’s group, who also created the calculation (Reference number 14 in the manuscript). They analysed the performance z-scores at different cut-offs (1, 1.5 and 2) on their ability to quantify and predict the risk of future disease activity. Future disease activity refers to the occurrence of relapse, EDSS worsening or EDA-3. This information is now included in the manuscript, page 9, line 217.

Comment 13 and 14 questions the correlation in Figures 4 and 5. We have found a moderate positive correlation (r= 0.447) between age and NfL concentration of the MS patients. According to many sources, including Boston University (The Correlation Coefficient (r) (bu.edu)) an r value between 0.4 and 0.5 is considered to have a moderate correlation.  A weak negative correlation (r= -0.125) has been found between BMI and NfL concentration of the MS patients. This has been rephrased to a ‘very weak negative correlation’ in the figure legend of figure 5 on page 6, line 142.

Comment 15 mentions that our statement of “Minor changes may be more easily detected using the Simoa method than the ELISA” is not shown in the study. That sentence has now been rephrased to indicate that this is an assumption we make based on the data we have gathered and the analytical sensitivity of Simoa. Further research with longitudinal samples can evaluate this hypothesis. Changes can be seen on page 10, line 257-259

Comment 16 questions whether serum NfL levels can predict the course of the progression of the disease and forecast upcoming MRI scans. CSF and blood concentration levels of NfL are highly correlated. NfL concentration increases in both in response neuroaxonal damge. As also stated in the reference supporting this statement on page 10 line 271.

Comment 17 “Serum was extracted and stored in aliquots of up to 1 ml at -20 ˚C” – Is there any data how freeze-thaw affects NfL concentration?

According to Marleen Koel-Simmelink et al., concentration of NfL in CSF were not negatively affected by repeated freeze-thaw cycles (up to 4).  (Marleen J A Koel-Simmelink, Anke Vennegoor, Joep Killestein, Marinus A Blankenstein, Niklas Norgren ,Carsten Korth, Charlotte E Teunissen, 2014, The impact of pre-analytical variables on the stability of neurofilament proteins in CSF, determined by a novel validated SinglePlex Luminex assay and ELISA, Journal of immunological methods, 402(1-2):43-9. doi: 10.1016/j.jim.2013.11.008).

We could not find any data for serum samples. Our samples, other than being stored in aliquots, were collected for the purpose of this experiment and were not thawed more than once before the quantification of NfL.

Comment 18 addresses the missing infomartion about the ELISA kit used. The reference number and country of manufacture has been added to the text, Page 11, line 303.

Comment 19, ‘5. Patents’ has been deleted from the manuscript.

Reviewer 2 Report

Your study is well-designed and carried out as well. Basically your work shows that two methodologies to assess NfL in serum in patients with MS provide similar results:  Simoa Advanced technology and the commercial available kit for ELISA. Your discussion on factors such age and BMI requiring to be adjusted in the interpretation of the results points to some of the hurdles analytical techniques are faced with in this regard. Some other factors not discussed may raise questions from the reader and this reviewer: 

1.  The patient of the MS clinical cohort you selected, Secondary Progressive (SPMS) form, for sample acquisition were practically all treated with a B-Cell depleting agent. (a) Readers would question the indication of this therapy in SPMS patients, but more importantly, (b) the audience needs assurance these patients had no detectable MRI or clinical activity which potentially may alter the levels of NfL, regardless the method utilized for quantification. 

2. You mention (lines 255, 256) Simoa is "more expensive" than the ELISA kit, but the reader is not provided with a quantifying idea of the difference (USD? Euros?). 

Author Response

Reviewer 2:

Comment 1 discusses the need to explain to the reader the effects of the B-cell depleting agents that all of our SPMS patients were treated with, as well as clarify that none of the MS patients had MRI detectable or going clinical activity, which may alter the levels of NfL. We would like to point out that treatment is solely a decision of the clinician following the patient at the time and this manuscript is based on the scientific data available.  This point has been discussed on page 10 lines 257-261. A limitation of our study is the lack of information on the clinical activity of the MS patients at time of sample collection. However, patients on B-cell depleting medication show stable level of NfL in serum, no detectable MRI activity or clinical relapses. (Reference number 21).

Comment 2 refers to the price difference between Simoa and ELISA as we have stated that the Simoa is more expensive. The prices for both methods have been specified on page 11, lines 286-289.

Round 2

Reviewer 1 Report

 The authors have answered all the reviewers’ comments and modified the manuscript accordingly. I have some minor suggestion which do not require a new round of peer review

1) page 9 line 188. The dynamic range of an ELISA is the range of antigen concentrations that can be accurately measured. Therefore, authors should add the measurement range provided by the manufacturer (4–100 pg/mL)

https://www.umandiagnostics.se/cms/wp-content/uploads/2022/01/Instructions-for-use-Serum-NF-light-RUO-v2022-01.pdf

2) Some objects in figures 3 and 7 appear to be blurry

Author Response

Reviewer 1

Comment 1, is regarding the dynamic range of the ELISA used by Uman Diagnostics. Page 9, line 188 now includes the dynamic range of the ELISA.

Comment 2, refers to Figure 3 and 7 appearing blurry. Both figures have now been replaced with a higher quality image.

Thanking you in advance.

Reviewer 2 Report

Your answers clarify this reviewer's questions. Thank you. 

Author Response

Thank you for reviewing the manuscript “A comparison of two analytical approaches for the quantification of neurofilament light chain, a biomarker of axonal damage in Multiple Sclerosis.” (Manuscript ID: ijms-2464051)